# The bearing capacity of asteroid (65803) Didymos estimated from boulder tracks

J. Bigot[1], P. Lombardo[1], N. Murdoch [1]✉, D. J. Scheeres [2], D. Vivet[1],
Y. Zhang [3], J. Sunshine [4], J. B. Vincent [5], O. S. Barnouin [6], C. M. Ernst [6],
R. T. Daly [6], C. Sunday[4], P. Michel [7], A. Campo-Bagatin [8], A. Lucchetti [9],
M. Pajola [9], A. S. Rivkin [6] & N. L. Chabot [6]

The bearing capacity - the ability of a surface to support applied loads - is an important parameter for understanding and predicting the response of a surface. Previous work has inferred the bearing capacity and trafficability of specific regions of the Moon using orbital imagery and measurements of the boulder tracks visible on its surface. Here, we estimate the bearing capacity of the surface of an asteroid for the first time using DART/DRACO images of suspected boulder tracks on the surface of asteroid (65803) Didymos. Given the extremely low surface gravity environment, special attention is paid to the underlying assumptions of the geotechnical approach. The detailed analysis of the boulder tracks indicates that the boulders move from high to low gravitational potential, and provides constraints on whether the boulders may have ended their surface motion by entering a ballistic phase. From the 9 tracks identified with sufficient resolution to estimate their dimensions, we find an average boulder track width and length of 8.9 ± 1.5 m and 51.6 ± 13.3 m, respectively. From the track widths, the mean bearing capacity of Didymos is estimated to be 70 N/m², implying that every 1 m² of Didymos' surface at the track location can support only ~70 N of force before experiencing general shear failure. This value is at least 3 orders of magnitude less than the bearing capacity of dry sand on Earth, or lunar regolith.

Geotechnical properties of asteroids affect their geology and evolution[1] and are important parameters in numerical models of the formation and history of small bodies. Moreover, they are also important for any space mission involving surface operations or interactions[2]. Direct measurements of the geotechnical properties made in the extremely low-gravity environment of the asteroid surface have the potential to inform the design of future space missions and instrumentation, and to reduce operational risk. One such geotechnical property is the ultimate bearing capacity or load bearing strength,

which corresponds to the maximum pressure that a surface can withstand without experiencing general shear failure[3,4]. The bearing capacity provides a means to determine if the surface of the considered celestial body is able to support the weight of a lander, rover, instrument or even an astronaut, and is also a potential measure for the trafficability of the surface material, i.e., whether the soil can provide traction and propulsion[5–8].

In preparation for the crewed Apollo missions, the geotechnical properties of the lunar soil were an important cause of concern[5]. On

¹Institut Supérieur de l'Aéronautique et de l'Espace (ISAE-SUPAERO), Université de Toulouse, Toulouse, France. ²University of Colorado, Boulder, CO, USA.
³Department of Climate and Space Sciences and Engineering, University of Michigan, Ann Arbor, MI, USA. ⁴University of Maryland, College Park, MD, USA.
⁵DLR, Cologne, Germany. ⁶Johns Hopkins Applied Physics Laboratory, Laurel, MD, USA. ⁷Université Côte d'Azur, Observatoire de la Côte d'Azur, CNRS,
Laboratoire Lagrange, Nice, France. ⁸University of Alicante, Alicante, Spain. ⁹INAF-OAPD Astronomical Observatory of Padova, Padova, Italy.
✉e-mail: naomi.murdoch@isae.fr

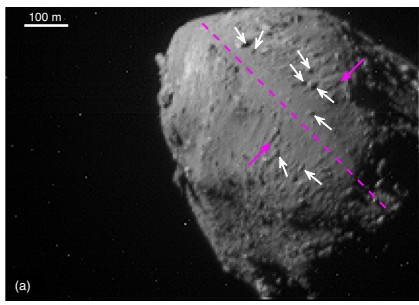
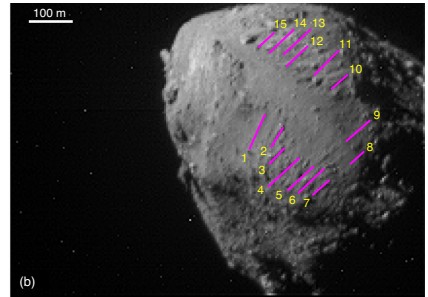

**Fig. 1 | Identification of suspected boulder tracks on asteroid Didymos. a** The approximate equator (dashed magenta line), example boulder tracks (magenta arrows) and likely boulders (white arrows) on the surface of Didymos. **b** The 15 boulder tracks identified on the surface of Didymos are indicated by the magenta lines. The image used here is a cropped section of DRACO image 22206, after Laplacian filtering.

Earth, the ultimate bearing capacity of a specific terrain can be deduced using in situ measurements such as plate loading tests (e.g., ref. 9) combined with the Terzaghi equation[10]. In lieu of being able to perform such experiments on the Moon prior to the Surveyor and Apollo missions, the load bearing strength of the lunar soil was derived from images of boulder tracks formed by rockfalls. Two major types of studies, with different assumptions, have been carried out, considering either the static boulder[7,8] or the rolling boulder[11] to compute the bearing capacity of the lunar soil from Lunar Orbiter photographs[7,8,10,11]. These studies found that the lunar surface load bearing strength ranges from approximately $10^2$ to $10^3$ kN/m$^2$.

More recently, these Apollo-era methods have been refined and applied to high-resolution imagery to determine whether different types of soils in the pyroclastic deposits and in the permanently shadowed regions of the Moon can be traversed by a vehicle[12,13]. Remote sensing images from the Lunar Reconnaissance Orbiter were used to determine the lunar bearing capacity using boulder track measurements[12,13] and the approach was adapted to be applicable to polar regions of the Moon and their illumination conditions[14]. Specifically, the Terzaghi geotechnical equation[12] gives the bearing capacity of the surface as a function of the local gravity, the soil properties (cohesion, internal friction angle, density), and the boulder track parameters (depth, width). Another approach is to use the Hansen equation which also depends on the slope and the boulder shape[12]. The values of bearing capacity derived from the boulder tracks[12] were found to correlate well with the known values of the highlands and mare regions provided by the Apollo missions[7,8,10,11].

On the 26th September 2022 (UTC), NASA's Double Asteroid Redirection Test (DART[15]) mission performed a kinetic impact into the 151-meter-size asteroid Dimorphos, the secondary asteroid orbiting around the 780-meter-diameter primary asteroid Didymos[16,17]. The DART impact reduced the orbit period of Dimorphos by 33 min[18], produced a large amount of ejecta[19], and was highly effective in deflecting the asteroid[20]. In the seconds before impact, DRACO (Didymos Reconnaissance and Asteroid Camera for Optical navigation[21]) took images of the binary system at a constant phase angle of ~59°[16,17]. These images showed, at the pixel scale of the available images, Didymos to have a relatively smooth equatorial region compared to the polar terrains. Linear groove-like features perpendicular to the equator can be seen, some of which appear to contain boulders (see Fig. 1).

Applying the estimated values for Didymos' size, mass, gravity field and spin rate[22] shows that the whole asteroid has a very small effective gravity ($g_{eff}$), reaching as low as $g_{eff} = 2.44 \times 10^{-5}$ m/s$^2$ in the equatorial region due to the centripetal acceleration ("Surface conditions on Didymos and lift-off speed limits" in "Methods"). The combination of the low gravitational acceleration and the centripetal acceleration that increases toward the equator facilitates the motion and transport of surface material, with a preferential direction of motion being from regions of high to low geopotential i.e., from the poles of the asteroid to the equator. In addition to having been observed directly on other asteroids such as Bennu[23,24], such surface motion has previously been proposed to explain the presence of Didymos' equatorial bulge[25], and to be at the origin of mass-shedding and even binary asteroid formation[26,27]. Based on these considerations, we assume that the linear features have been formed by rolling or sliding boulders on a granular surface. Indeed, as a boulder moves downhill it pushes away and compacts the regolith material resulting in the formation of a groove[28]. The linear features on Didymos are similar in appearance to tracks observed on both the Moon (e.g., refs. 7,8,10–12) and Mars (e.g., refs. 29–31), which have been attributed to boulder motion on steeply sloping surfaces[32]. Tracks formed by bouncing boulders have also been observed on comet 67P[33]. On the surface of Didymos, the linear tracks are all parallel and directed toward the minimum gravitational potential found at the equator (Figs. 1 and 2). The tracks do not appear to show an increasing width, indicating that boulder motion is more likely to have caused the features than general avalanching or mass wasting that may be associated with a debris apron. The boulder tracks also occur in the region of Didymos identified to have the least stable surface according to a "factor of safety" slope stability analysis (ratio of resisting frictional and cohesive stresses to gravity[17]), and some tracks appear to have a terminal boulder (Fig. 1).

In this work, we first identify and measure boulder tracks on the surface of Didymos. We then apply a previously validated geotechnical approach[12] to estimate the bearing capacity of the asteroid surface. Given the extremely low-gravity environment, we also investigate in detail the validity of the underlying assumptions in the geotechnical analysis and address the likelihood of boulders lifting-off the surface at the terminus of the boulder tracks.

## Results

### Boulder tracks

We consider the last complete image of Didymos taken with DRACO before the DART impact (image 22206 from the PDS—see "Data availability"). Fifteen possible boulder tracks are identified on the surface of Didymos, of which 9 are chosen to be studied further. We apply a Laplacian filter to enhance the image contours and facilitate track measurements ("Image processing" in "Methods") then we manually measure the width ($B$) and length ($L$) of each track ("Boulder track analyses" in "Methods"). The track widths range from 6.6–11.5 m (1.5–2.6 pixels) with a mean and standard deviation of 8.9 ± 1.5 m. The track lengths range from 32.3–74.4 m (7.3–16.8 pixels) with a mean and standard deviation of 51.6 ± 13.3 m. The image filtering has a minimal (-2%) influence on the track measurements ("Influence of filtering and resolution" in "Methods"). The relative pixel scale of the image (4.43 m) with respect to the track dimensions means that we are at the limit of resolution. We estimate that we have a measurement error of up to ~20% in the width measurements,

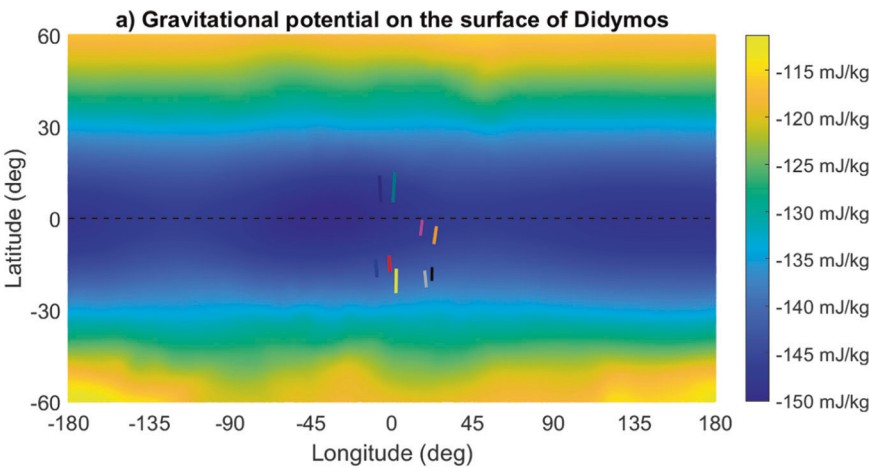

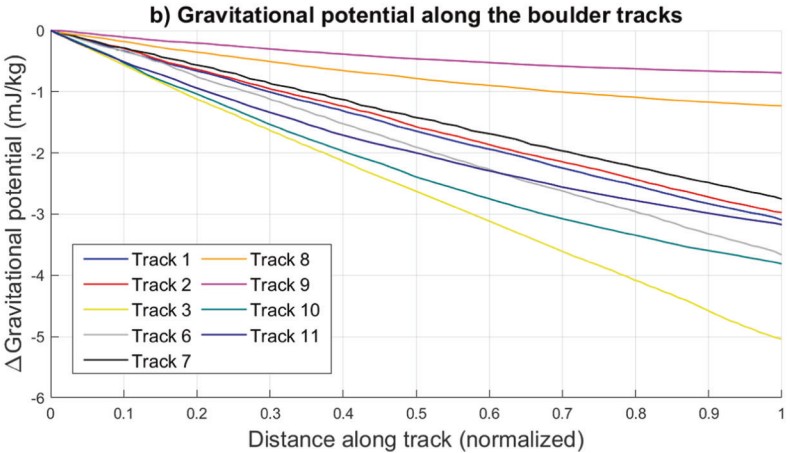

**Fig. 2 | Topography of Didymos at the boulder track locations. a** Map of the gravitational potential over the surface of the asteroid Didymos from ref. 17 with the positions of the identified boulder tracks (Fig. 1) indicated. **b** The change in gravitational potential along each of the tracks is shown as a function of the normalized track length (i.e., along track distance/total track length) measured from higher to lower latitudes.

of this we attribute ~10% to the influence of the image resolution ("Influence of filtering and resolution" in "Methods"). The bearing capacity estimate takes into account a possible error of one pixel (~50% error) on the width measurements to address this uncertainty.

The detailed topographic data from ref. 17 provides evidence for a change in the gravitational potential on the surface of Didymos. The surface presents a minimum of gravitational potential at the equator (Fig. 2 and Supplementary Fig. 1) due to the fast rotation of Didymos, supporting the hypothesis of the observed lineaments being tracks formed by boulders moving from higher to lower latitudes. The boulder tracks are located on either side of the equator, between −24° and 16° in latitude ("Boulder track analyses" in "Methods"). We note that the analysis of a rolling or sliding boulder subject to friction is not expected to lead to a significant change in longitude, consistent with the boulder track observations being at a fixed meridian ("Surface conditions on Didymos and lift-off speed limits" in "Methods").

The change in gravitational potential along each of the selected tracks is reported in Fig. 2. All 9 boulder tracks exhibit a lower gravitational potential at the lower latitude end of the track (toward the equator) indicating that the boulders would have moved from the higher to lower latitudes in order to minimize their gravitational potential energy.

**The bearing capacity of Didymos' surface**
We use the measurements of tracks formed by boulders moving on granular surfaces to infer the bearing capacity of the surface. This

method has been used to measure the bearing capacity and trafficability of certain regions of the Moon[11–13]. Here, we use our derived measurements of the width ($B$) of boulder tracks on Didymos to estimate the bearing capacity of Didymos' surface via the Terzaghi equation ("Estimating the bearing capacity of Didymos' surface" in "Methods"). This equation, commonly used in geotechnical engineering for the determination of the shear strength of soil, provides the maximum load that the surface material is able to sustain before general shear failure. At this failure state, the shear strength of the underlying ground balances the weight of the boulder. As a result, the properties of the boulder can be neglected and only the surface properties are considered[12].

To provide an uncertainty in the mean bearing capacity that reflects the possible ranges of the key parameters ($D$, $B$, $\phi$, $c$, $\rho$, GM, $R$), we perform a Monte Carlo simulation. As it is not possible to determine the track depth ($D$) from the DRACO images (previous lunar studies have used shadowing effects but the illumination conditions and viewing geometry do not allow for this in the DRACO images), we consider the track depth to be a free variable in our analyses that can range between zero and the diameter of the boulder, i.e., between zero and the track width: $D \in [0;B]$. To be extremely conservative about our track width measurement errors, we allow the track width ($B$) to vary in the range of $\bar{B} \pm 4.43$ m (1 pixel), where $\bar{B}$ is the mean track width (8.9 m). The angle of internal friction ($\phi$) is varied from 25° to 45°, since this is the expected range for geological materials[5]. The cohesion is varied from 0 to 10 N/m² in order to include the extreme case of no

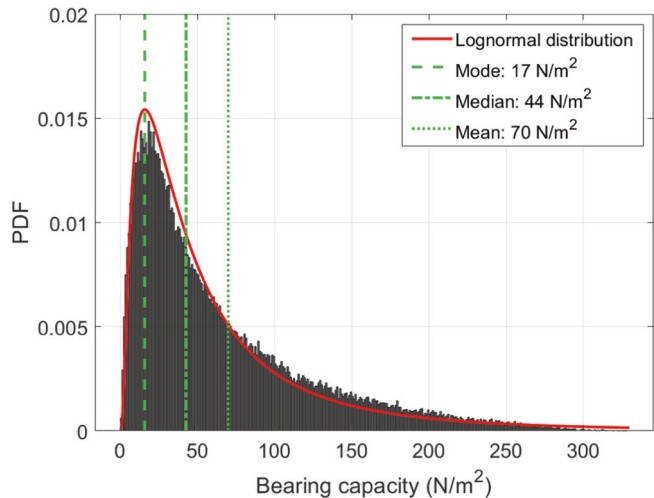

**Fig. 3 | The bearing capacity of the surface of Didymos estimated using a Monte Carlo simulation.** Uniform distributions are assumed for the key variables in the range of: $\phi \in [25;45]°$, $c \in [0;10]$ N/m², $\rho \in [1000;3000]$ kg/m³, GM $\in [33.9; 36.9]$ m³/s², $R \in [383; 405]$ m, $B \in [4.47; 13.33]$ m, $D \in [0; 13.33]$ m. A total of 100,000 iterations are performed and the histogram of the results is shown in gray. The fitted lognormal distribution (shown in red) has a mean and mode of 70 and 17 N/m², respectively. The lognormal fit has parameters $\mu = 3.8$ and $\sigma = 1.0$.

cohesion[17] and the pre-encounter estimates of 10 N/m² based on a study of the stability of Didymos[34]. The radius of Didymos is varied from 383 m to 405 m and the gravitational acceleration at the mean latitude of 13° is varied from $2.3 \times 10^{-4}$ m/s² to $2.8 \times 10^{-4}$ m/s² corresponding to the proposed possible range of asteroid radius R and GM parameters (394 ± 11 m and 35.4 ± 1.5 m³/s² respectively[22]) and taking into account the oblateness of the asteroid ("Surface conditions on Didymos and lift-off speed limits" in "Methods"). These variations lead to the net surface acceleration varying in the range of $1 \times 10^{-6}$ m/s² to $6.3 \times 10^{-5}$ m/s² at the mean track latitude of 13°. We call this the effective gravitational acceleration ($g_{eff}$). Finally, we also vary the regolith density from 1000 to 3000 kg/m³.

The results at each track depth are fitted by a lognormal distribution (Fig. 3) giving a mean bearing capacity for the surface of Didymos of 70 N/m² (mode of 17 N/m², 95% Confidence Intervals: 7 to 190 N/m²). This distribution of bearing capacity values reflects the variations of track-to-track widths and the possible ranges of the key parameters. The bearing capacity remains below 300 N/m² even for the most extreme combinations of parameters (95% confidence intervals: 7 to 190 N/m²). The ~20% measurement uncertainty on the track widths ("Influence of filtering and resolution" in "Methods") is small with respect to the uncertainties introduced by the poorly-constrained parameters.

### Influence of cohesion, friction, regolith density and local gravity on the bearing capacity

Here we perform a parametric study to quantify the influence of the poorly constrained parameters on our results for the bearing capacity of the surface of Didymos. We first establish a set of baseline parameters for Didymos. The regolith bulk density is estimated to be 2790 ± 140 kg/m³ (equivalent to the estimated bulk density for Didymos[22]). The angle of internal friction is assumed to be 35° compatible with the geology of Didymos[17,35] and the morphology of boulders on Dimorphos[36]. A baseline value of 1 N/m² is assumed for the cohesion of the surface material on Didymos[17]. The radius and GM parameter estimates for Didymos are 394 ± 1 m and 35.4 ± 1.5 m³/s², respectively[22], leading to an estimated gravitational acceleration of $2.54 \times 10^{-4}$ m/s² and a net surface acceleration ($g_{eff}$) of $3.1 \times 10^{-5}$ m/s² at the mean track latitude (13°) taking into account the gravitational

acceleration, shape and the rotational acceleration of Didymos (assuming the current spin period of 2.26 h[37], see "Surface conditions on Didymos and lift-off speed limits" in "Methods"). The density of the surface material is assumed to be the same as the bulk density. Assuming these baseline parameters ($\phi = 35°$, $c = 1$ N/m², $\rho = 2790$ kg/m³, $g_{eff} = 3.1 \times 10^{-5}$ m/s²) the bearing capacity of the surface of Didymos is estimated to be between 12 and 23 N/m² over the range of the unknown track depth ($D \in [0;B]$; Supplementary Fig. 9).

From the sensitivity analyses (Fig. 4) we can see that the bearing capacity increases with increased friction, increased cohesion, increased density and increased gravity. The higher these values, the greater the resistance of the soil to shear forces and therefore, the greater the bearing capacity. We highlight that the cohesion has the largest influence on the bearing capacity, followed by the angle of friction. On the contrary, the gravitational acceleration and the regolith density have minimal influences on the resulting bearing capacity within the ranges tested. With regards to the track width, a very conservative 1-pixel error (~50% error) on the width measurements leads to an uncertainty of ± 1.5 N/m² on the bearing capacity estimation (assuming the baseline parameters provided above).

### Typical boulder speeds and lift-off criterion

For a boulder to form a track, it has to be in contact with the surface. A boulder of diameter 9 m (equivalent to the mean track width) will lift off the surface of Didymos (R = 394 m) at a speed of 0.097 m/s at the equator and at 0.11 m/s at 13° of latitude, corresponding to an angular speed of about 0.024 rad/s ("Surface conditions on Didymos and lift-off speed limits" and "Estimating the speed of boulders" in "Methods"). Although the track widths are expected to be nearly independent of both the boulder speed and the local slope angle[38], it is nonetheless interesting to consider the speeds at which boulders may move across the surface of Didymos under the influence of gravity to determine if they will reach this lift-off criterion. It is not known whether the boulders were rolling or sliding when they formed the tracks. To estimate the boulders speeds in the rolling case, the boulders are considered as rolling spheres on an inclined plane covered with a granular material, and we use the mechanical model[38] that takes into account the penetration depth, the gravitational acceleration and the local inclination ("Estimating the speed of boulders" in "Methods", Supplementary Fig. 10). To estimate the speed that rolling boulders can reach when creating the boulder tracks on the surface of Didymos we assume a simplified calculation whereby a boulder starts rolling from zero velocity and rolls toward the equator on a constant slope and is subjected to a constant gravitational acceleration ("Estimating the speed of boulders" in "Methods"). The assumption of boulders starting from rest is most compatible with the geological and dynamic evolution of the binary system: the spin-up of the asteroid leads to the motion of surface material from pole to equator (e.g., refs. 17,23,26). As we cannot distinguish the boulders themselves in order to measure their radii (r), we assume that the diameter of the boulder that formed the track is equal to the width of the trench (i.e., B = 2r). Assuming a local slope of 45° corresponding to the mean gravitational slope found along the tracks (Fig. 2 and ref. 17), and a track depth of r/4 = 1.125 m, a rolling boulder of radius r = 4.5 m boulder would take about 5 h to travel a distance of around 600 m (approximately the distance from the pole of Didymos to the equator) reaching an angular (linear) maximum speed of 0.018 rad/s (0.08 m/s).

The topographic data provided by ref. 17 gives the average local gravitational slope along each of the individual boulder tracks. We use this, combined with the net acceleration at the mid-latitude of each track, to estimate the terminal speeds of the boulders (Supplementary Fig. 12). The largest velocity reached by a rolling boulder is 0.007 rad/s (0.03 m/s) (track 10, Supplementary Fig. 12). In the case of a sliding boulder, the linear velocity will depend on the sliding friction force between the boulder and the regolith-covered surface ("Estimating the

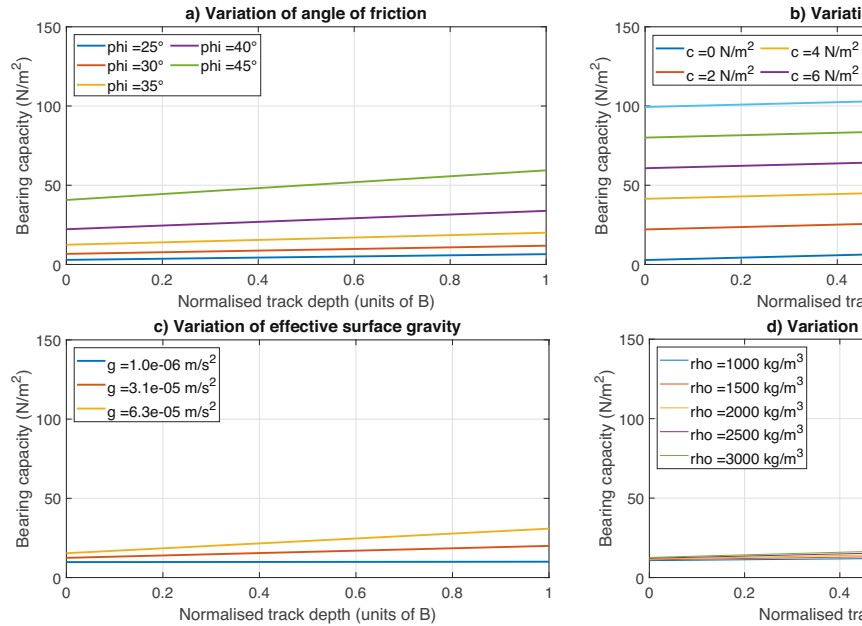

**Fig. 4 | The influence of uncertain parameters.** In these figures the bearing capacity assuming the mean track width (8.9 m) and the mean latitude (13°) is plotted as a function of normalized track depth. The parameters are varied individually while keeping the others equal to the baseline values ($\phi = 35°$, $c = 1$ N/m², $g_{eff} = 3.1 \times 10^{-5}$ m/s², $\rho = 2790$ kg/m³). **a** Influence of varying the angle of internal friction from 25 to 45°. **b** Influence of varying the cohesion from 0 to 10 N/m². **c** Influence of varying the effective gravitational acceleration ($g_{eff}$) from $1 \times 10^{-6}$ to $6.3 \times 10^{-5}$ m/s², corresponding to a GM range of $35.4 \pm 1.5$ m³/s² and $R = 394 \pm 11$ m. **d** Influence of varying the regolith density from 1000 to 3000 kg/m³.

speed of boulders" in "Methods"). Assuming the extreme case of zero friction, the maximum velocity reached by a boulder sliding along the longest track (track 10) is 0.06 m/s ("Estimating the speed of boulders" in "Methods"). These estimated speeds from both the rolling and sliding analyses are below the lift-off speed of 0.024 rad/s (0.11 m/s) for such a boulder implying that the lift-off speed was not reached by the boulders. Therefore, with the current best estimate of the local effective gravity, boulders forming these tracks are not expected to have reached the lift-off velocity ("Surface conditions on Didymos and lift-off speed limits" in "Methods"). The boulders may, therefore, have stopped due to a change in local surface topography as they approach the equator (Fig. 2, refs. 17,[25]).

**Validity of underlying assumptions**

The geotechnical approach, with the use of the Terzaghi equation, is applied with several assumptions. Unlike the images of asteroid Dimorphos[35], the images of Didymos do not offer enough resolution to distinguish the track-forming boulders themselves and their real shapes[36]. The Terzaghi equation assumes that the boulders are spherical, creating circular footprints in the granular soil, and that the ground is a level surface with respect to the gravity vector. Other geotechnical models exist that take into account the local slope and boulder shape when calculating the bearing capacity i.e., the Hansen model[39] that was also used to analyze lunar boulder tracks in ref. 12. Although gravitational slope data are available[17], the boulder shapes are not known. Therefore, we use the more simplified Terzaghi equation, which can be considered to provide an upper bound to the bearing strength[12].

The Terzaghi approach can be applied in an identical manner both in the case of rolling or sliding boulders. However, an important implicit assumption is that the boulders are moving slow enough for the boulder−regolith interactions to be considered as quasi-static. Specifically, granular materials such as regolith can exhibit solid (quasi-static), liquid (hydrodynamic) and gas-like (dilute) behaviors. As a granular material in different regimes does not exhibit the same response (e.g., refs. 40,41), it is important to determine which regime the surface material of Didymos was in when the tracks formed in

order to determine whether a quasi-static approach for interpreting the surface response is applicable. The extremely low net surface acceleration leads to very slow boulder motion ("Surface conditions on Didymos and lift-off speed limits" and "Estimating the speed of boulders" in "Methods"). At such speeds, there would be little doubt in terrestrial or even lunar gravity that the interactions occurred in a quasi-static regime. However, in a very low-gravity environment a granular surface can become fluidized much easier due to the reduced frictional interactions[40,42]. When fluidized, the surface−boulder interactions occur in the inertial regime and the quasi-static hypothesis no longer holds.

To assess the validity of the quasi-static hypothesis, we study the regime of the boulder−surface interactions by means of the rotational Froude number, $Fr_\omega = \frac{\omega^2 r}{g_{eff}}$, with $\omega$ being the angular speed of the boulder, $r$ the boulder's radius and $g_{eff}$ the net acceleration at the surface. The Froude number at the limit between the quasi-static and inertial regimes can be defined as the transition Froude number and is likely to be in the region of 1–10[41]. However, without a detailed study, it is unclear exactly where the transition Froude number is situated. This value will also vary according to the material properties[41]; increased friction and cohesion means increased quasi-static behavior and consequently a larger transition Froude number.

If we assume a transition Froude number of 5, a 9 m diameter boulder rolling down a 45° slope would switch to the inertial regime after rolling over around 70 m (Supplementary Fig. 11). In other words, the underlying Terzaghi assumption of quasi-staticity is valid over the first ~70 m of rolling. This value is independent of the value of the gravitational acceleration and depends only on the local slope and the boulder size (Supplementary Figs. 11 and 13). The maximum angular speeds estimated at the end point of 7 out of 9 tracks ("Estimating the speed of boulders" in "Methods", Supplementary Fig. 12) are not large enough for a quasi-static to inertial regime transition to have occurred (assuming a transition Froude number of 5). The boulders forming the other two tracks may have switched to the inertial regime before they stop, but for the major part of the formation of these tracks, and for all other tracks, the quasi-static assumption is valid.

## Discussion

The linear groove-like features perpendicular to the equator on Didymos seem to be tracks formed by boulders moving down the gravitational slope from higher to lower latitudes in order to reduce their gravitational potential energy. This conclusion has been reached using the gravitational potential derived from the current best estimates of the shape model and density of Didymos[17]. However, we note that these may change slightly with updated measurements from the upcoming Hera mission[43].

We measure 9 of the tracks and find an average track width of 8.9 ± 1.5 m and an average track length of 51.6 ± 13.3 m. Then, using the same technique that has been used to infer the bearing capacity of lunar soil using boulder tracks visible on the surface of the Moon[12,13], we estimate the mean bearing capacity of Didymos to be 70 N/m² (mode of 17 N/m², 95% confidence intervals: 7–190 N/m²). This would mean that every 1 m² of Didymos' surface, in the region of the tracks, can support only ~70 N of force before experiencing failure. This is at least 1000 times smaller than the bearing capacity of dry sand on Earth (~10⁵ N/m²; [IS:1904-1988][44]), and 5000 times smaller than the bearing capacity of lunar regolith[4,5,11,12].

Given the extremely low surface gravity environment (~10⁻⁵ m/s² at the mean track latitude of 13°), special attention is paid to the underlying assumptions of the geotechnical approach. Specifically, we ensure that contact is maintained for long enough to form the tracks, i.e., the boulders do not lift off, and that the surface interactions likely occur in the quasi-static regime thus confirming the validity of implicit assumption of the Terzaghi equation. In our analyses of the boulder motion, we assume that the boulders start from rest, likely triggered by surface instabilities due to the fast rotation of the asteroid. Other plausible triggers for boulder motion include stress variations due to tidal forces from the secondary asteroid (Dimorphos) or impact, thermal or tidally induced seismicity (e.g., refs. 45,46). If the initial boulder velocities are larger this could result in the boulder motion moving out of the quasi-static regime invalidating the use of the Terzaghi equation. However, the initial velocity must also be lower than the lift-off velocity; if the initial boulder velocity is close to or above the lift-off velocity then the boulders would lose contact with the surface before forming the tracks.

The estimation of bearing capacity depends on the shear mode (general, local, or punching shear) experienced by the material underneath the boulders. The shear regime depends on the relative density of the soil and on the angle of internal friction[14,47]. In our analyses we assumed a general shear mode, which means a larger portion of the soil is activated during failure[48]. Based on the conclusions of refs. 14,47, this is a reasonable hypothesis given the assumed regolith bulk density (2790 kg/m³), and likely high (~35°) angle of internal friction[17,35,36]. However, as the shear mode also depends on the depth of the track, which is not well known, other shear modes could contribute. As the general shear mode leads to a higher bearing capacity estimate than other regimes[48], the estimates of bearing capacity provided here should be considered an upper bound to the bearing strength.

The estimated weak load bearing strength of the asteroid Didymos is compatible with estimations made of the very weak tensile strength (<100 Pa) of comets[49,50] and is also consistent with recent observations of very weak surface material on asteroids Bennu and Ryugu[51,52]. The OSIRIS-REx spacecraft collected a sample of the surface of asteroid Bennu with a robotic arm named TAGSAM (Touch-and-Go Sample Acquisition Mechanism)[24,53]. The collision speed ($v_c$) of TAGSAM with the asteroid surface was ~0.1 m/s and the surface gravity ($g$) on Bennu is equal to $6 \times 10^{-5}$ m/s²[51]. The TAGSAM diameter ($d$) is 0.305 m[54]. The penetration depth of TAGSAM in Bennu's surface (before release of the gas) was ~0.064 m[51]. Using the surface properties of asteroid Bennu (friction angle 30°–40°, cohesion ≪ 1 N/m²[24,51]), and

using the same geotechnical analyses as above (here in Eq. (2), $B = d$ and $D$ is the TAGSAM penetration depth), the estimated bearing capacity of Bennu is of the order of 1–20 N/m², a value that is in the same order of magnitude than the one found here for Didymos. However, the collisional Froude number at the impact[55] is $Fr_c = \frac{v_c}{\sqrt{gd}}$ where here the length scale is the TAGSAM diameter ($d$). This gives a Froude number of impact of 23, far above the predicted transition Froude number of 1.5 for impacts[55]. Therefore, the TAGSAM interaction with Bennu is likely to have occurred in the inertial regime[55], in which case the quasi-static Terzaghi approach is not appropriate.

The Hera mission[43] will provide high-resolution (0.5–2 m/px) images of the global surface of Didymos. This will provide additional information and potential confirmation that the observed linear features are indeed tracks formed by boulders. The Hera images will lead to improved measurements of the tracks investigated here, and perhaps also of the associated boulders in some cases. In particular, observations at different phase angles and the improved shape model will allow the track depths to be further constrained in turn reducing the uncertainties on the bearing capacity estimates. As tracks may be submitted to erosion over time, their shapes may be modified: older tracks become wider as they are degraded leading to a less reliable estimate of the bearing capacity[12,56]. Only recent tracks are good indicators for mechanical behavior of a surface. The age of the tracks is not discussed in this paper given the limitations of the image resolution but it will be investigated with the additional data provided by the Hera mission[43]. The DART impact generated a large volume of ejecta[19], some of which may have remained in the Didymos system and reimpacted either Didymos or Dimorphos[57] at low velocity. If this is the case, the reimpacting boulders may have generated additional tracks that can be studied using data from the Hera mission[43]. In addition, the Hera CubeSats, Milani and Juventas, plan to land on the surface of the secondary asteroid Dimorphos. The landing dynamics will be measured by on-board accelerometers and these measurements can provide additional data to constrain the response and physical properties of asteroid surfaces[58] at even lower gravity levels.

## Methods

Here we describe (1) the image processing, (2) the boulder track measurements, (3) the influence of filtering and resolution on the track measurements, and (4) an analysis of the speed and lift-off criteria of boulders on the surface of Didymos.

### Image processing

The DART spacecraft carried a panchromatic, narrow angle, visible imager called DRACO (Didymos Reconnaissance and Asteroid Camera for Optical navigation)[21]. DRACO imaged both asteroids during the approach to Dimorphos. To make our main measurements, we used an image with a pixel scale of 4.43 m (image 22206 from the PDS—see "Data availability"). However, to examine the influence of the filtering and resolution on our measurements ("Influence of filtering and resolution" in "Methods") we also used two other images that contained parts of Didymos (images 10933 and 05363 from the PDS—see "Data availability") with pixel scales of 3.47 m and 3.38 m, respectively.

To visualize more clearly and then measure the boulder tracks ("Boulder track analyses" in "Methods") on the surface, image processing solutions were used to enhance high spatial frequencies and zones with high intensity gradients which reflect the details and edges that we want to emphasize. The Laplacian filter increases the visibility of the contours of objects by identifying and enhancing the zones with high intensity variations. More specifically, the Laplacian filter detects contours by searching for the maximum intensity gradients corresponding to a zero second derivative, which reflect discontinuities in intensity or edges. It is a convolution filter, meaning that it multiplies each pixel of the image matrix by a kernel. This operation highlights

the zones of high gradient where there is a fast variation of intensity. Different kernels exist to achieve the objective of increasing the contours with a more or less strong sharpening, and the kernel $K$ we have selected, presented hereafter, gives the best resolution and contrast for our study.

$$K = \begin{pmatrix} 0 & 1 & 0 \\ 1 & -4 & 1 \\ 0 & 1 & 0 \end{pmatrix} \qquad (1)$$

This simple kernel enables the intensity to be detected in all directions, as it gives equal importance to horizontal, vertical and diagonal directions, a feature that may vary in other kernels that focus on one direction.

A High Boost Filtering method can also be used for amplifying high frequencies corresponding to high detail areas. However, we chose to use the Laplacian filter because it produced better results than the High Boost Filtering and presented the most useful surface details for identifying and measuring boulder tracks. An example raw image taken by DRACO on DART, and the same image after the High Boost filter and the Laplacian filter are shown in Supplementary Fig. 2. This is one of the last images with Didymos still visible entirely, here in the bottom left corner, before Didymos left the DRACO field of view as the spacecraft approached the satellite Dimorphos. The boulder tracks become more visible thanks to the edge sharpening and we can observe several parallel lines oriented from the poles to the equator, as well as a wide smooth zone at the equator.

### Boulder track analyses

From the sharpened image ("Image processing" in "Methods"), we identified the boulder tracks and manually measured their widths in pixels by placing markers along the tracks and measuring the distance between the edges. The pixel scale is provided by the Small Body Mapping Tool (SBMT)[59]. This software allows the users to visualize and manipulate small body shape models in three dimensions while providing easy access to data from a variety of missions. We identified 15 possible tracks that are highlighted in Fig. 1. Among these tracks, 9 were clear enough to measure the width and length (Table 1). The track widths range from 6.6–11.5 m (1.5–2.6 pixels) with a mean value and standard deviation of 8.9 ± 1.5 m. The track lengths range from 32.3–74.4 m (6.9–15.9 pixels) with a mean value and standard deviation of 51.6 ± 13.3 m.

The coordinates in latitude and longitude of these 9 tracks were measured from the images using the Small Body Mapping Tool[59]. The boulder tracks are located on either side of the equator, between −24° and 16° in latitude, and oriented from pole to equator. The elevation and gravitational potential maps of Didymos' surface as well as the location of the tracks are shown in Fig. 2. Supplementary Fig. 1 shows that the gravitational potential, taken at the mean longitude of the tracks, is minimum at the equator. This creates a gravitational slope along which boulders would tend to move toward the potential minimum. These data, as well as the dynamic model of Didymos implying that its rapid spin creates a centripetal acceleration driving material to the equator, support the hypothesis of the observed lineaments being tracks formed by rolling boulders driven by gravitational slopes.

### Influence of filtering and resolution

Image filtering is necessary to enhance the visibility of the tracks. However, to quantify the influence of the image filtering and resolution on the boulder tracks measurements, we use two partial images of Didymos (images 10933 and 05363 from the PDS—see "Data availability") with pixel scales of 3.47 m and 3.38 m, respectively. In these partial images only tracks 1, 2, and 3 fully are visible (see Supplementary Fig. 3). We measure the widths of these tracks both in the raw and the filtered images and report the values in Supplementary Table 1. We

**Table 1 | Width and length measurements of nine boulder tracks, measured manually on image 22206 (pixel scale of 4.43 m) enhanced with a Laplacian filter ("Image processing" in "Methods")**

| Track number | Width (in m) | Length (in m) | Start latitude (°) | End latitude (°) | Start longitude (°) | End longitude (°) |
|---|---|---|---|---|---|---|
| 1 | 8.4 | 46.2 | −19 | −14 | −8 | −9 |
| 2 | 9.9 | 53.7 | −18 | −12 | −1 | −2 |
| 3 | 10.2 | 64.1 | −24 | −17 | 2 | 2 |
| 6 | 7.8 | 35.3 | −23 | −17 | 19 | 18 |
| 7 | 6.6 | 32.3 | −21 | −16 | 22 | 22 |
| 8 | 7.9 | 54.7 | −8 | −3 | 23 | 25 |
| 9 | 9.3 | 46.5 | −6 | −1 | 16 | 17 |
| 10 | 11.5 | 74.4 | 15 | 6 | 2 | 1 |
| 11 | 8.9 | 57.1 | 14 | 5 | −7 | −6 |

Tracks 4, 5, 12, 13, 14 and 15 (Fig. 1) were not of sufficient resolution to be measured. The latitudes and longitudes at the start, middle and end of each track were measured with the Small Body Mapping Tool (SBMT)[59].

also compare the widths of these three tracks as measured in the baseline image for our analyses (image 22206), which has a pixel scale of 4.43 m. The filtering (comparison of the measurements made on the raw images vs. enhanced images) is found to have a minimal influence on the track width measurements (average error of 2.2%, max error of 6.5%; Supplementary Table 1).

To quantify the possible measurement error introduced due to image resolution, we also compare the widths of the same track measured in images with different resolutions (comparison of the measurements made on the enhanced images 22206, 10933, 05363 in Supplementary Table 2). The difference in the track widths measured in different images with different resolutions can be as large as -20%. Given that these are not the same image, some of these differences could also be attributed to slightly varying viewing geometry and/or illumination conditions between the images rather than only resolution. However, the geometry was constant during the encounter and the illumination conditions changed only minimally.

Nonetheless, in order to isolate the effects of the image resolution, we took the finest resolution (image 05363) image containing part of Didymos and subsequently gradually degraded the resolution. The pixel scale was degraded in intervals of 0.5 m (from 3.38 m to 7.88 m) and the degraded images are shown in Supplementary Fig. 4. As the pixel scale is degraded, artefacts appear and the tracks become less visible. When the resolution reaches -1 pixel per track width (corresponding to 40% of the original resolution) the tracks cannot be measured (with or without image filtering). For each degraded image in which the tracks were visible, we measured the widths and lengths of the three tracks (Supplementary Tables 3 and 4). The percentage error in measured track width with respect to the width measured in the original image increases as the resolution degrades (Supplementary Fig. 5). For a pixel scale of 4.43 m (as in our main image in which we have measured the 9 tracks), the error due in the width and length measurements due to the image resolution is -5% and -10%, respectively.

To further improve the estimations of the error in the width measurements we also took a high-resolution image of lunar boulder tracks and degraded the resolution. To have the optimal comparison between the DRACO image of Didymos and the lunar image, we selected a Lunar Reconnaissance Orbiter Camera image (image ID M135215829RC) with the same phase angle (60°) as DRACO image 22206 (Supplementary Fig. 6a). Supplementary Fig. 6 shows the track and an example of the track width measurement. The lunar image with different resolution is shown in Supplementary Fig. 7. The error in the track width with respect to the original image is found to remain below 10% (Supplementary Fig. 8).

In conclusion, with respect to the track widths (the key measured parameter for determining the bearing capacity) we have a measurement error of up to ~20% due to the different appearance of tracks in the three different DRACO images analyzed, of this we attribute ~10% to the influence of the image resolution. We can also conclude that the image filtering improves the manual track detection without adding artefacts.

## Estimating the bearing capacity of Didymos' surface

We use the measurements of tracks left by boulders moving on granular surfaces to infer the bearing capacity of the surface. This method has been used to measure the bearing capacity and trafficability of certain regions of the Moon[11–13]. Here, we use our derived measurements of the width ($B$) of boulder tracks on Didymos to estimate the bearing capacity of Didymos' surface via the Terzaghi Eq. (2). The ultimate bearing capacity formula[3], which plays a role in geotechnical engineering for the determination of the shear strength of soil foundations, is given by the following:

$$q_f = 1.3cN_c + D\gamma_s N_q + 0.3\gamma_s B N_\gamma \qquad (2)$$

where $q_f$ is the bearing capacity of the surface (N/m²), $D$ is the track depth (m), $B$ is the track width (m), $c$ is the cohesion of the surface material (N/m²), $\gamma_s = \rho g$ is the unit weight of homogeneous soil (N/m³). In the case of Didymos, the gravitational acceleration ($g$) is replaced by the effective gravitational acceleration ($g_{eff}$), that takes into account both the gravitational acceleration and the rotational acceleration of Didymos ("Surface conditions on Didymos and lift-off speed limits" in "Methods"). The $N$ factors of the equation are defined as follows:

$$N_q = \frac{e^{\left(\frac{3\pi}{4} - \frac{\phi}{2}\right)\tan(\phi)}}{2\cos^2\left(45° + \frac{\phi}{2}\right)} \qquad (3)$$

$$N_c = (N_q - 1) - \cot(\phi) \qquad (4)$$

$$N_\gamma = \frac{2(N_q + 1)\tan(\phi)}{1 + 0.4\sin(4\phi)} \qquad (5)$$

where $\phi$ is the internal friction angle. The three terms of this equation represent the total shear strength of the soil with the contributions of the cohesion, the soil surcharge, and the angle of internal friction, respectively. This equation provides the ultimate bearing capacity of the surface, which is the maximum load that the ground is able to sustain before general shear failure. At this failure state, the shear strength of the underlying ground balances the weight of the boulder. As a result, the properties of the boulder can be neglected and only the surface properties are considered[12].

## Surface conditions on Didymos and lift-off speed limits

If Didymos is modeled as a rotating sphere locally about the equator, a simple model for surface gravity conditions can be found. The key parameters are found in ref. 22 and are the Didymos gravitational parameter of $GM = 35.4$ m³/s² and an equatorial radius of $R = 394$ m. This gives a surface gravitational attraction of $g = 2.280 \times 10^{-4}$ m/s². The spin period of Didymos is 2.260 h, or a spin rate of $\omega = 7.723 \times 10^{-4}$ rad/s. This translates to a centripetal acceleration at the equator of $2.350 \times 10^{-4}$ m/s², which is notable as it is greater than the gravitational attraction. If we also add the effect of the asteroid oblateness, using the estimated $J2 = 0.09$ gravity coefficient, we find that the gravity is increased by 13.5%, increasing the equatorial gravity to $g = 2.588 \times 10^{-4}$ m/s², and yielding a downward effective gravity of $g_{eff} = 2.384 \times 10^{-5}$ m/s². The effective gravity will vary as a function of the latitude $\delta$ around the equator, due to changes in the centripetal

acceleration and the oblateness contribution to the attraction. At a latitude of 13° the effective surface gravity is $3.1 \times 10^{-5}$ m/s².

For this simple model the effective gravity attraction can be computed using the formula:

$$g_{eff} = \frac{GM}{R^2}\left[1 + 0.135\left(3\cos(\delta)^2 - 2\right)\right] - R\omega^2\cos(\delta)^2 \qquad (6)$$

If a boulder has enough speed, it can also achieve ballistic lift-off from the surface[60]. The condition is found by computing the surface normal force due to gravity, rotation of the asteroid, and additional speed of the particle accounting for the curved shape of the asteroid surface. When the additional speed is along a meridian, the speed limit is computed as $v \geq \sqrt{g_{eff}R}$ where R is the asteroid radius. At a latitude of 13°, we see that the lift-off speed is 0.11 m/s while at the equator the speed limit is 0.097 m/s. If the latitude at which the boulder trail ends is measured, it may place a constraint on the speed on the boulder if indeed it experienced lift-off. However, we note that the effect of the boulder center of mass being above the surface can also lead to lift-off[61,62].

These speed limits can be related to the Froude number given a boulder radius. If we assume a boulder with radius $r = 4.5$ m at a latitude of 13°, the Froude number is $F_R = v / \sqrt{g_{eff}r} < 9.1$ given the speed limit. This corresponds to a rolling rate of 0.014 rad/s or less. A simple analysis of the effect of Coriolis accelerations at these slow speeds shows that they will be small in general, consistent with the grooves appearing to lie along meridians of the asteroid.

## Estimating the speed of boulders

The maximum speed ($v$) that a boulder within a certain time ($t$) can obtain while rolling on the surface of Didymos is estimated assuming a simplified calculation whereby the boulders are considered as rolling spheres on an inclined plane covered with a granular material (Supplementary Fig. 10). A boulder of mass $m$ and radius $r$ starts rolling from zero initial velocity ($v_0$) and rolls toward the equator on a constant slope ($\alpha$), subjected to a constant gravitational acceleration ($g$) and experiencing a friction force $F_d$ i.e.:

$$v(t) = gt\sin(\alpha) - \frac{F_d}{m}t + v_0 \qquad (7)$$

The mechanical model presented in ref. 38 allows us to estimate the friction force between the boulder and the granular surface taking into account both the penetration depth and the local gravity in order to calculate the rotational velocity. Equation (8) represents the friction force ($Fd$) to which a boulder is subjected, coming from its sinkage in the granular surface. The sinkage is expressed using the angle $\theta_m$ formed by the zone of contact between the boulder and the grains, which itself depends on the boulder radius $r$ and the penetration depth $D$ of the boulder in the granular medium (Eq. (9)). The angle $\theta_c$ is defined from the orientation of the normal and tangential stresses with respect to the slope[38], expressed as a function of $\theta_m$ and of an empirical constant $k = 1.5$ adjusted from experimental data (Eq. (10)):

$$F_d = mg\cos(\alpha)\left(\frac{\frac{2}{5}\tan(\alpha) + \sin(\theta_c)}{\frac{2}{5} + \cos(\theta_c)}\right) \qquad (8)$$

$$\cos\theta_m = 1 - \frac{D}{r} \qquad (9)$$

$$\theta_c = \theta_m\left(1 - e^{-\frac{k\alpha}{\theta_m}}\right) \qquad (10)$$

When considering Didymos we replace $g$ in the above equations by the effective gravity ($g_{eff}$) to account for the rotational acceleration in addition to the gravitational acceleration.

We vary successively the parameters of the slope, of track depth (penetration depth of the boulder in the granular surface) and of gravity to assess their respective influence. The angular velocity of a boulder as a function of its displacement is shown in Supplementary Fig. 11. A given boulder will only start to move steadily beyond a critical slope[38]. The value of this critical slope depends on the local gravitational acceleration as can be seen in Supplementary Fig. 11a; the slope is varied from 0° to 50° but for low slopes of 0°–20°, the boulder does not move. The velocity reached by the boulders depends also on the depth at which they are buried in the surface, as this influences the frictional interactions[38]. Indeed, it can be seen in Supplementary Fig. 11b in which we vary the penetration (track) depth between 0 and 4 m, that the greater the penetration (track) depth, the slower the boulder rolls down the slope. The mean gravitational slope along the tracks, using the data from ref. 17, is 45°. For this gravitational slope, and baseline value of effective gravity ($3.1 \times 10^{-5}$ m/s²), the boulder cannot start moving when the depth becomes greater than 2 m, indicating that the critical slope angle depends not only on the local gravitational acceleration, but also on the penetration depth. Finally, we also consider the resulting angular speed for a range of values of gravitational accelerations (accounting also for the influence of the rotation and oblateness of Didymos). We choose to consider the range of $1 \times 10^{-6}$ m/s² to $6.3 \times 10^{-5}$ m/s² for the effective gravity ($g_{eff}$) at the mean latitude (13°), corresponding to the proposed possible range of $GM$ (35.4 ± 1.5 m³/s²) and $R$ (394 ± 11 m) reported in ref. 22. For a distance of 52 m (the mean track length), a 9 m diameter rolling boulder will only reach speeds of 0.009 rad/s (0.04 m/s), In the case of a sliding boulder, the linear velocity will depend on the sliding friction force between the boulder and the regolith-covered surface. Assuming the extreme case of zero friction, a 9 m diameter boulder will reach a speed of 0.045 m/s after sliding a distance of 52 m (Supplementary Fig. 11d). These estimated speeds from both the rolling and sliding analyses are below the lift-off speed of 0.024 rad/s (0.11 m/s) for such a boulder. For illustrative purposes, on Supplementary Fig. 11 we show the angular velocities for a 9 m diameter boulder corresponding to rotational Froude numbers ($Fr_{\omega} = \frac{\omega^2 r}{g_{eff}}$, with $\omega$ being the angular speed of the boulder, $r$ the boulder's radius and $g_{eff}$ the effective gravity at the surface) of 1, 5 and 10.

We also apply this model to the specific values of gravitational slope and latitude found for the 9 boulder tracks, provided by the topographic data of ref. 17. The local gravitational slope of each track and its mean latitude taken into account for estimating the net surface acceleration are used to determine the speeds reached by the boulders forming these tracks. We assume a 4.5 m radius ($r$) boulder and a baseline track depth of $r/4 = 1.125$ m. Supplementary Fig. 12 shows that these speeds do not exceed the limit of regime transition for 7 tracks out of 9. The quasi-static assumption is valid for most of the tracks and for the major part of the other two tracks.

For the above analysis, we assumed that the boulders are rolling without sliding. If in fact the boulders formed the tracks by sliding and not rolling then, the linear velocity will depend on the sliding friction force $F_s$ between the boulder and the regolith-covered surface. This force is equal to the product between the dynamic friction coefficient and the weight leading to the following linear velocity[38]:

$$v(t) = gt\sin(\alpha) - \frac{F_s}{m}t + v_0 \quad (11)$$

$$\text{With}: F_s = \mu_D mg\cos(\alpha) \quad (12)$$

Again, we replace $g$ in the above equations by the effective gravity $g_{eff}$ (Eq. (6)) in order to account for the rotational acceleration of Didymos in addition to the gravitational acceleration.

The linear velocity of a sliding boulder, for different dynamic friction coefficients, as a function of its displacement is shown in Supplementary Fig. 11d. The linear velocity decreases as the dynamic friction coefficient increases to such an extent that, above a certain value of this coefficient, the boulder no longer moves. With these simplified models presented above, the lift-off speed ("Surface conditions on Didymos and lift-off speed limits" in "Methods") is only reached in a very small number of cases (very steep slopes, large boulders, larger effective gravity and long displacements). This is shown by the red dashed line in Supplementary Fig. 11.

Again, using the topographic data of ref. 17, the largest anticipated sliding velocity (corresponding to track 10, with a mean gravitational slope and latitude of 53° and 10°, respectively) in the extreme case of no friction is 0.06 m/s.

Supplementary Fig. 13 evaluates the influence of the latitude on the lift off speed, the regime transition speed and the angular speed of the 9 m boulder. Here the effect of the asteroid's rotation is taken into account. We assume the current rotation rate of Didymos (rotation period $T = 2.26$ h) for this exercise, although we note that the tracks may have formed during a period of time when the rotation rate was different. It can be seen that the lift off speed and the regime transition speed decrease with decreasing latitude, but the angular speed of the boulder also decreases. The distance traveled before transitioning from the quasi-static to the inertial regime, however, remains constant with latitude (Supplementary Fig. 13d).

## Data availability
The DART DRACO data are publicly available in the Planetary Data System. The figures used (images "dart_0401929905_22206", "dart_0401929940_05363", "dart_0401929937_10933", referred to as images "22206", "05263" and "10933" in this paper) can be found on the PDS (Planetary Defense System) website (https://pds-smallbodies. astro.umd.edu/holdings/pds4-dart:data_dracocal-v2.0/SUPPORT/ dataset.shtml); Small Bodies Node; DART Calibrated Images for the Didymos Reconnaissance and Asteroid Camera for OpNav (DRACO) instrument v2.0; Browse; "Final" Parent Directory. Source data and images are provided with this paper and can be accessed here: https:// doi.org/10.5281/zenodo.11109757.

## Code availability
The codes used to perform the analyses and generate the figures of this paper are available here: https://doi.org/10.5281/zenodo.11109757.

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

## Acknowledgements

This work was supported by the DART mission, NASA Contract No. 80MSFC20D0004. N.M. acknowledges funding support from the European Commission's Horizon 2020 research and innovation programme under grant agreement No. 870377 (NEO-MAPP project) and support from the French Space Agency (Centre National d'Etudes Spatiales; CNES), focussed on the Hera space mission. This work was supported by the Italian Space Agency (ASI) within the LICIACube project (ASI-INAF agreement no. 2019-31-HH.0) and HERA project (ASI-INAF agreement no. 2022-8-HH.0). P.M. acknowledges support from the French Space Agency (CNES) and ESA.

## Author contributions

J.B. and P.L. contributed equally to this work performing all of the data analysis, geotechnical interpretations and supporting calculations in this paper in addition to making important contributions to the writing of the manuscript. They should be considered as joint first authors. N.M. was the supervising author, led the writing of the manuscript and guided the analysis and interpretations. D.S. provided the dynamical analysis. D.V. contributed to the image analysis. Y.Z., J.S., and J.B.V. helped to significantly improve the quality of the analysis. O.S.B., R.T.D., and C.M.E. helped develop foundational data products, including DRACO image calibration. C.S. provided the initial work focussing on dynamical regimes and the Froude number, in addition to contributing comments on this work. P.M., A.C.-B., A.L., and M.P. all helped to improve the quality of the manuscript. A.S.R. and N.L.C. provided a leadership role in the DART mission.

## Competing interests

The authors declare no competing interests.
