## [Peer Review File · Nature Communications]

The bearing capacity of asteroid (65803) Didymos estimated from boulder tracksREVIEWER COMMENTS

Reviewer #1 (Remarks to the Author):

Dear authors,

thank you for submitting this highly interesting and relevant work. I carefully went through the manuscript and believe that it deserves prompt publication, once a handful of major and minor issues have been addressed. This review includes a PDF file with a total of 54 comments. My main concerns are summarized below.

1) Boulder tracks – or not?

The entire manuscript is built on the assumption that the observed lineaments are boulder tracks. However, there is not much presentation/discussion of evidence that supports that fundamental assumption. Does the topography of Didymos enable the displacement of boulders? How are the boulder tracks located/oriented in relation to the topography? Are there boulder source regions? Not all tracks feature visible boulders – how can you be certain that boulders are the cause of the lineaments? Did some of the boulders disintegrate after their displacement? Were they covered by impact ejecta? What triggered the displacement of the boulders in the first place? How did the trigger influence the velocity of the boulders? As one example, a figure that showcases Didymos' topography as e.g. presented in Barnouin et al. would go a long way, especially if combined with a schematic cross-section of the surface along a boulder track. On another note, the age of the tracks and the impact on the estimates of bearing capacity are not discussed; older tracks are wider (e.g. through edge erosion and diffusive softening of the tracks' edges by micro-meteorites etc.) which can affect the measurements of track width and thus of bearing capacity.

2) How many pixels does one need for a reasonable measurement of track width?

One of the most important parameters in the used equations is the diameter of the circular footing, i.e., the measurement of the track width. I have two concerns:

a. As shown by past work, e.g. on the Moon, the width of boulder tracks can wildly fluctuate along the track, due to irregularities of the boulder, topography, etc.; past work imperfectly addressed that issue by deriving a number of track width measurements along the same track, then using an average width for the bearing capacity estimates. If possible, this work could enhance the robustness of its results by implementing a similar approach.

b. After reading the manuscript and looking at all figures, I am not convinced that track width can be reasonably well measured in the available images. The tracks on Didymos are less than 2 pixels to ~3 pixels wide (in the used image 22206). In other words, the edges of a track are constrained by roughly ~two adjacent pixels. How do you recognize the edges of the tracks, i.e., where exactly do you measure track width? I am attaching an example of a lunar boulder track below, in full resolution as well as down-sampled to 4.43 m. It is challenging enough to accurately measure the width of tracks on the Moon, using images that resolve track width with tens of pixels. The down-sampling experiment you performed is a solid first step in verifying your measurements, but it is limited to other poorly resolved images. I would suggest running another experiment using e.g. lunar images, measuring track width in full resolution as ground truth, then down-sampling to lower resolutions (as done in the manuscript). Ideally, the person making the measurements would start measuring track width using the lowest resolution image(s) first, which would make sure that there is no knowledge transfer (if the high-res images are measured first, the person performing the measurements could subconsciously learn where the edges of the tracks are located in the images, skewing all subsequent measurements). Such an experiment would go a long way in building trust in the results of this work, especially if visualized with an additional figure showing example measurements, etc.

SEE FIGURE IN ATTACHED .DOCX

Figure – ~8 m-wide boulder track on the Moon, image ID M113934119L

(https://wms.lroc.asu.edu/lroc/view_lroc/LRO-L-LROC-3-CDR-V1.0/M113934119LC). Left, full resolution (0.5 m), right, down-sampled to 4.42 m.

3) Assumption of a general shear regime

The used equations assume that the soil/regolith fails in a general shear regime, but there is no

discussion of why that assumption was made. Depending on the soil's physical properties and the environment (also dependent on the depth of the track) the soil might fail in local or punching shear, which would require a modification of the used equations (and would result in lower bearing capacity estimates). In Bickel & Kring (2020) we discussed that issue with an eye on the lunar regolith; on the Moon, the first few ~cm are expected to fail in punching and local shear while deeper soil is expected to fail in general shear.

4) Terzaghi vs. Hansen

The most recent work utilizing boulder tracks on the Moon (as referenced by your manuscript) is using Hansen's updated bearing capacity equations, as they allow for some fine-tuning to local slope, etc. (using correction factors). I understand why you have chosen to fall back to the original Terzaghi equations, but there is no clear description/discussion in the manuscript (which should be added).

5) Communication of uncertainties

The estimation of bearing capacity using orbital observations of boulder tracks is affected by a number of uncertainties, even if high-resolution images and some local information about the soil's physical properties are available (like on the Moon). This work does not have any ground truth for Didymos' physical properties – also, there are no high-resolution images. The uncertainties are thoroughly discussed, but somewhat separated from the presentation of the main results. Personally, I would advise to add a brief mention of the significant uncertainties to all sections that present results, to make sure a reader understands that the presented results are not definitive. One of the motivations of this manuscript is the determination of engineering parameters; given this work is providing an "upper bound" of bearing capacity only, the communication of uncertainties is particularly important.

6) Methods

There are quite a few sections and paragraphs in the main text that should be moved to the dedicated Methods section, per Nature Communications' requirements.

I hope you find my comments and suggestions useful!
Thank you and best regards – and a happy new year,
Valentin Bickel

Reviewer #2 (Remarks to the Author):

In the study "The bearing capacity of asteroid (65803) Didymos estimated from boulder tracks" Bigot et al. have calculated the bearing capacity of the asteroid Didymos using the Terghazi formula which depends on the boulder track depth, material strength, and gravitational acceleration. They have also estimated the effect of various parameters on the bearing capacity, and examined the validity of the Terghazi equation assumptions. The bearing capacity of any planetary body determines the load it can carry without yielding or collapsing. It is necessary to know before any lander mission to the body. The study is significant in the view of upcoming lander missions and, to understand the origin and evolution of asteroids. Sufficient details are provided to repeat the study and corroborate the results and analysis.

Major comment: As the foundation of the present study is based on the identification and measurement of the boulder tracks width and length, it is necessary to be sure of the accuracy of these measurements. The authors have claimed to identify and measure the boulder tracks in the range 1.5-2.6 pixels which, according to Pajola et al., 2019, are below the 3-pixel resolution requirement from the Nyquist theorem. It is true that the high phase angle (~59° in the present case) casts a long shadow and it becomes easier to identify the surface undulations (or boulders), but the resolution in this study does not seem to be sufficient to measure the tracks. The error in the measurement could be much larger (if the identified grooves are indeed the boulder tracks) than estimated in the study. The authors have degraded the original image (number-05363, resolution-3.38 m) to different resolutions from 3.88 m to 7.88 and estimated the percent error

which comes out to be in the range of 5-10% (maximum of ~20%). The authors are already starting with the assumptions that they are able to rightly measure the track widths in the range 6.6-11.5 m, but I would have liked to see the authors begin with boulders tracks > 6m width (from NAC LROC images) at the similar phase angles to the DRACO images and then degrade those to the DRACO resolution. This would show that these tracks look similar, and that the authors can really measure these boulder tracks from the DRACO images. Most of the other analysis that follows the track width and length measurement can then be built upon it.

Minor comments:

Sup Fig 2: "tracks 1, 2, and 3 are indicated with the orange arrows", the arrows look purple to me!

Sup Fig 4: Why should the measurement error decrease when degrading to a lower resolution e.g., in Track 2 between 5 m and 6 m, and Track 3 between 5.5 m and 6.5 m?

Sup Fig 8: Can't see the separate dotted/dashed lines. I only see 7 lines which can either be dashed or dotted.

Methods

1) Use either GM or μ throughout the text.

14/03/2024

We would like to thank the reviewers for their very helpful and constructive comments that significantly improved the quality of this paper. Our detailed responses are shown in blue below and we have also addressed all of the comments/suggestions contained with the annotated pdf. In the revised manuscript the new or modified text is highlighted in yellow and we have also clearly indicated (in purple) the numerical values that have changed since the first submission.

Kind regards,

Jeanne Bigot, Pauline Lombardo, Naomi Murdoch and coauthors

1st reviewer

1) Boulder tracks – or not?

The entire manuscript is built on the assumption that the observed lineaments are boulder tracks. However, there is not much presentation/discussion of evidence that supports that fundamental assumption. Does the topography of Didymos enable the displacement of boulders? How are the boulder tracks located/oriented in relation to the topography?

Thank you for this comment. We now have the detailed topographic data available (from Barnouin et al., 2024) and have considered in detail the change in elevation and gravitational potential of the surface of the asteroid in the specific locations of the tracks. This has confirmed that indeed the displacement of boulders is possible. This also allows us to demonstrate exactly how the boulder tracks are oriented with respect to the local topography. In addition, it can be seen that the tracks with the largest change in elevation between the top and the bottom are also the longest; consistent with a larger boulder velocity moving further on the surface. This information is summarized in the new figures added to the paper and they are also included here for reference:

Topography of Didymos at the boulder tracks locations. (a) The surface elevation and (c) gravitational potential are represented all over the surface of the asteroid. The positions of the boulder tracks are also shown on the maps. (b) change in surface elevation and (d) change in gravitational potential along the tracks, with distance along track normalised with respect to the track length and measured from higher to lower latitudes. The variations of elevation and gravitational potential are shown with respect to their values at the starting point of the track.

Gravitational potential versus latitude at mean longitude of the boulder tracks. The region of boulder tracks is defined by the maximum latitudes found for the tracks.

Are there boulder source regions?

From this study we are not able to identify specific source regions although the identified tracks all begin in a latitude range of 6 to 24° (the mean latitude for the start of the track is 16°).

Not all tracks feature visible boulders – how can you be certain that boulders are the cause of the lineaments?

The asteroid Didymos has several particularities with respect to larger planetary bodies such as the Moon. This asteroid is rotating relatively fast (rotational period of 2.26 hrs) and is very small (~780 m). The combination of these two factors facilitates the motion and transport of surface material with a preferential direction of motion being from regions of high to low geopotential i.e. from the poles to the equator. Such surface motion has previously been proposed to explain the origin of binary asteroids (e.g., Walsh et al., 2008), in addition to having been observed directly on other asteroids such as Bennu (Jawin et al. 2020, Barnouin et al., 2022). Therefore, it is entirely plausible that there is motion of boulders from higher to lower latitudes. Our analysis of the local elevation and geopotential changes at the locations of each of the tracks (see figures above) also confirms that such motion is entirely possible at the specific track locations. Of course, with the limited data we have available we cannot be entirely sure that the lineaments are due to boulder tracks, but from all of the information we have available, it does appear to be the most likely cause. We have added some extra justification for our hypotheses into the paper introduction.

Barnouin, O. S., Daly, M. G., Seabrook, J. A., Zhang, Y., Thuillet, F., Michel, P., et al. (2022). The formation of terraces on asteroid (101955) Bennu. Journal of Geophysical Research: Planets, 127, e2021JE006927. <https://doi.org/10.1029/2021JE006927>

Jawin, E. R., Walsh, K. J., Barnouin, O. S., McCoy, T. J., Ballouz, R.-L., DellaGiustina, D. N., et al. (2020). Global patterns of recent mass movement on asteroid (101955) Bennu. Journal of Geophysical Research: Planets, 125, e2020JE006475. <https://doi.org/10.1029/2020JE006475>

*Walsh, K., Richardson, D. & Michel, P. Rotational breakup as the origin of small binary asteroids. Nature **454**, 188–191 (2008). <https://doi.org/10.1038/nature07078>*

Did some of the boulders disintegrate after their displacement?

Unfortunately, within the limits of the resolution we are not able to determine if any of the boulders disintegrated. However, we note that the local gravity is very small and the typical displacement speeds are < 5 cm/s (Sup. Fig. 13) so the boulders are not likely to have experienced a large force during their motion. The Hera mission will allow us to study the boulders themselves in more detail.

Were they covered by impact ejecta?

What triggered the displacement of the boulders in the first place?

We apologize but we are not entirely sure what impact ejecta the reviewer is referring to here. That said, we do not expect that the boulder motion was triggered by a boulder fall (e.g. from a cliff), as is often the case on the Moon and Mars. Here, we expect the trigger to be unstable material on the surface moving to a lower geopotential (as mentioned above, this is not uncommon on small fast rotating asteroids). In addition to surface instabilities due to fast rotation, other plausible triggers include stress variations due to tidal forces from the secondary asteroid (Dimorphos), or impact, thermal or tidally induced seismicity (e.g., Murdoch et al., 2017, DellaGiustina et al., 2024). However, if there is impact ejecta associated with the boulder motion, there is no evidence for this in the images. However, it is entirely possible that the tracks have subsequently been covered by impact ejecta from the DART impact. Again, the Hera mission will allow us to study this.

Both the triggering and the possible re-impacting debris from the DART impact are now discussed in the Discussion section.

DellaGiustina, et al. Seismology of rubble-pile asteroids in binary systems, Monthly Notices of the Royal Astronomical Society, 2024,; stae325, <https://doi.org/10.1093/mnras/stae325>

Murdoch, et al. Probing the internal structure of the asteroid Didymos with a passive seismic investigation, Planetary and Space Science, 144, 2017, Pages 89-105, <https://doi.org/10.1016/j.pss.2017.05.005>.

Why are there boulders at all?

Boulders exist on the surfaces of all asteroids visited by space missions. In fact, many small asteroids are actually considered to be rubble pile asteroids meaning they are made entirely of boulders rather than being monolithic material. This is a consequence of having experienced a catastrophic disruption event i.e., an impact from another asteroid that completely broke the asteroid to pieces. It is generally accepted that these pieces then reaccumulated together under gravitational forces to form a gravitational aggregate made of rubble (e.g., Farniella et al., 1982, Holsapple and Housen et al., 2019), hence the name rubble pile. Any boulders on the surface of Didymos may either be due to such a catastrophic disruption process, or they may have been formed in-situ by non-catastrophic impacts.

Farinella, Paolicchi, Zappalà, The asteroids as outcomes of catastrophic collisions, Icarus, Volume 52, Issue 3, 1982, Pages 409-433, [https://doi.org/10.1016/0019-1035\(82\)90003-3](https://doi.org/10.1016/0019-1035(82)90003-3).

Holsapple and Housen, The catastrophic disruptions of asteroids: History, features, new constraints and interpretations, Planetary and Space Science, Volume 179, 2019, 104724, <https://doi.org/10.1016/j.pss.2019.104724>.

How did the trigger influence the velocity of the boulders?

In this study we assume that the boulders start from rest, as this is most compatible with the geological and dynamic evolution of the binary system (spin up of the asteroid leading to pole -> equator motion of surface material). However, if another mechanism triggered the

displacement, the velocity may have been slightly higher at the start of the motion. Without detailed knowledge of the trigger process, we cannot evaluate the consequences of this.

As one example, a figure that showcases Didymos' topography as e.g. presented in Barnouin et al. would go a long way, especially if combined with a schematic cross-section of the surface along a boulder track.

Thank you, this was a great suggestion. The data from Barnouin et al. (2024) is now available and we have made figures of the elevation and gravitational potential of the surface as a function of distance along the track (see above).

On another note, the age of the tracks and the impact on the estimates of bearing capacity are not discussed; older tracks are wider (e.g. through edge erosion and diffusive softening of the tracks' edges by micro-meteorites etc.) which can affect the measurements of track width and thus of bearing capacity.

This is also a great point but it is not possible for us to investigate it here given the constraints with the image resolution. We have, however, added some text to mention this point in Discussion.

2) How many pixels does one need for a reasonable measurement of track width?

One of the most important parameters in the used equations is the diameter of the circular footing, i.e., the measurement of the track width. I have two concerns:

a. As shown by past work, e.g. on the Moon, the width of boulder tracks can wildly fluctuate along the track, due to irregularities of the boulder, topography, etc.; past work imperfectly addressed that issue by deriving a number of track width measurements along the same track, then using an average width for the bearing capacity estimates. If possible, this work could enhance the robustness of its results by implementing a similar approach.

b. After reading the manuscript and looking at all figures, I am not convinced that track width can be reasonably well measured in the available images. The tracks on Didymos are less than 2 pixels to ~3 pixels wide (in the used image 22206). In other words, the edges of a track are constrained by roughly ~two adjacent pixels. How do you recognize the edges of the tracks, i.e., where exactly do you measure track width?

I am attaching an example of a lunar boulder track below, in full resolution as well as down-sampled to 4.43 m. It is challenging enough to accurately measure the width of tracks on the Moon, using images that resolve track width with tens of pixels. The down-sampling experiment you performed is a solid first step in verifying your measurements, but it is limited to other poorly resolved images. I would suggest running another experiment using e.g. lunar images, measuring track width in full resolution as ground truth, then down-sampling to lower resolutions (as done in the manuscript). Ideally, the person making the measurements would start measuring track width using the lowest resolution image(s) first, which would make sure that there is no knowledge transfer (if the high-res images are measured first, the person performing the measurements could subconsciously learn where the edges of the tracks are located in the images, skewing all subsequent measurements). Such an experiment would go

a long way in building trust in the results of this work, especially is visualized with an additional figure showing example measurements, etc.

SEE FIGURE IN ATTACHED .DOCX

Figure – ~8 m-wide boulder track on the Moon, image ID M113934119L (https://wms.lroc.asu.edu/lroc/view_lroc/LRO-L-LROC-3-CDR-V1.0/M113934119LC). Left, full resolution (0.5 m), right, down-sampled to 4.42 m.

These are really good suggestions and we thank the reviewer for taking the time to provide example lunar images for us to work with.

With the image resolution we currently have, making width measurements at different locations on the Didymos tracks does not make sense, unfortunately. However, we recognise the reviewers' concerns about the importance of the track width measurement and the associated uncertainties. We have therefore performed several additional analyses to better quantify the measurement uncertainty and also to understand the consequences of the associated uncertainty.

First, we have analysed degraded high resolution lunar track images as suggested (see more details below). We have followed the reviewer's advice and measured track widths at two different locations on a lunar track before degrading the images to a lower resolution (down to 5 m). To have a better comparison between the image of Didymos and the one of the Moon, we took a lunar image with the same phase angle as image 22206. The phase angle of the image of Didymos is 60° so, on the Lunar Reconnaissance Orbiter Camera website, we selected image ID M135215829RC and identified a track on it. We also now show an example lunar image with the track edges marked in order to provide an example of the width measurement (see below). Concretely, we placed markers along the tracks and measured the distance between the two edges with the Matlab tool "ImTool". Finally, we calculate the error of the track width measurement in each of the degraded images with respect to the original image and we find that the error, even in the worst case, is below 10%. This is the same methodology we applied to the image 05363 on Didymos, in which we found an error below 30%. These analyses are all described in Methods 3 and the associated figures.

Studied track on the Moon with an example of the track width measurement

Secondly, in the Monte Carlo analysis, to be extremely conservative about our track width measurement errors, we now assume a very conservative 1 pixel error (~50% error) on the track width measurements. For information, this leads to an uncertainty of +/- 1.5 N/m² on the bearing capacity estimation (assuming the baseline parameters provided in the paper). See the figure below.

Influence of track width measurement error on the bearing capacity. The estimated bearing capacity as a function of track depth assuming the baseline parameters and the mean measured track width of 8.9 m (red). In blue the track width is the mean track width - 1 px and in orange the track width is the mean track width + 1 px (1 px = 4.43 m).

3) Assumption of a general shear regime

The used equations assume that the soil/regolith fails in a general shear regime, but there is no discussion of why that assumption was made. Depending on the soil's physical properties and the environment (also dependent on the depth of the track) the soil might fail in local or punching shear, which would require a modification of the used equations (and would result in lower bearing capacity estimates). In Bickel & Kring (2020) we discussed that issue with an eye on the lunar regolith; on the Moon, the first few ~cm are expected to fail in punching and local shear while deeper soil is expected to fail in general shear.

Thank you for pointing this out. We have now added the following paragraph to the discussion:

“The estimation of bearing capacity depends on the shear mode (general, local, or punching shear) experienced by the material underneath the boulders. The shear regime depends on the relative density of the soil and on the angle of internal friction [Bickel and Kring 2020, Vesic, 1973]. In our analyses we assumed a general shear mode, which means a larger

portion of the soil is activated during failure [⁵⁹Costes et al., 1970]. Based on the conclusions of [¹⁴Bickel and Kring 2020,⁴¹Vesic, 1973], this is a reasonable hypothesis given the assumed regolith bulk density (2790 kg/m³), and likely high (~35°) angle of internal friction [⁵⁶Barnouin et al., 2024, ³⁶Pajola et al., 2023, ³²Robin et al. 2023]. However, as the shear mode also depends on the depth of the track, which is not well known, other shear modes could contribute. As the general shear mode leads to a higher bearing capacity estimate than other regimes [⁵⁹Costes et al., 1970], the estimates of bearing capacity provided here should be considered an upper bound to the bearing strength.”

4) Terzaghi vs. Hansen

The most recent work utilizing boulder tracks on the Moon (as referenced by your manuscript) is using Hansen’s updated bearing capacity equations, as they allow for some fine-tuning to local slope, etc. (using correction factors). I understand why you have chosen to fall back to the original Terzaghi equations, but there is no clear description/discussion in the manuscript (which should be added).

We have now added the following information to the **‘Validity of underlying assumptions’ section:**

“Other geotechnical models exist that take into account the local slope and boulder shape when calculating the bearing capacity i.e., the Hansen model [³⁷Hansen 1970] that was also used to analyse lunar boulder tracks in ¹²Bickel et al. 2019. Although slope data are available [⁵⁶Barnouin et al. 2024], the boulder shapes are not known. Therefore, we use the more simplified Terzaghi equation, which can be considered to provide an upper bound to the bearing strength [¹²Bickel et al. 2019].”

5) Communication of uncertainties

The estimation of bearing capacity using orbital observations of boulder tracks is affected by a number of uncertainties, even if high-resolution images and some local information about the soil’s physical properties are available (like on the Moon). This work does not have any ground truth for Didymos’ physical properties – also, there are no high-resolution images. The uncertainties are thoroughly discussed, but somewhat separated from the presentation of the main results. Personally, I would advise to add a brief mention of the significant uncertainties to all sections that present results, to make sure a reader understands that the presented results are not definitive. One of the motivations of this manuscript is the determination of engineering parameters; given this work is providing an “upper bound” of bearing capacity only, the communication of uncertainties is particularly important.

Thank you for this good advice. We now introduce the parameter uncertainties much earlier in the paper and have moved the Monte Carlo results earlier too. We also removed the ‘baseline’ bearing capacity values that were in the original manuscript to avoid confusion with the Monte Carlo results. We also explicitly state in the discussion that this should be considered to be an upper bound.

6) Methods

There are quite a few sections and paragraphs in the main text that should be moved to the dedicated Methods section, per Nature Communications' requirements.

We have now moved all of the geotechnical equations and associated methodology to the Methods section.

2nd reviewer

In the study "The bearing capacity of asteroid (65803) Didymos estimated from boulder tracks" Bigot et al. have calculated the bearing capacity of the asteroid Didymos using the Terghazi formula which depends on the boulder track depth, material strength, and gravitational acceleration. They have also estimated the effect of various parameters on the bearing capacity, and examined the validity of the Terghazi equation assumptions. The bearing capacity of any planetary body determines the load it can carry without yielding or collapsing. It is necessary to know before any lander mission to the body. The study is significant in the view of upcoming lander missions and, to understand the origin and evolution of asteroids. Sufficient details are provided to repeat the study and corroborate the results and analysis.

Major comment: As the foundation of the present study is based on the identification and measurement of the boulder tracks width and length, it is necessary to be sure of the accuracy of these measurements. The authors have claimed to identify and measure the boulder tracks in the range 1.5-2.6 pixels which, according to Pajola et al., 2019, are below the 3-pixel resolution requirement from the Nyquist theorem. It is true that the high phase angle ($\sim 59^\circ$ in the present case) casts a long shadow and it becomes easier to identify the surface undulations (or boulders), but the resolution in this study does not seem to be sufficient to measure the tracks. The error in the measurement could be much larger (if the identified grooves are indeed the boulder tracks) than estimated in the study. The authors have degraded the original image (number-05363, resolution-3.38 m) to different resolutions from 3.88 m to 7.88 and estimated the percent error which comes out to be in the range of 5-10% (maximum of $\sim 20\%$). The authors are already starting with the assumptions that they are able to rightly measure the track widths in the range 6.6-11.5 m, but I would have liked to see the authors begin with boulders tracks $> 6\text{m}$ width (from NAC LROC images) at the similar phase angles to the DRACO images and then degrade those to the DRACO resolution. This would show that these tracks look similar, and that the authors can really measure these boulder tracks from the DRACO images. Most of the other analysis that follows the track width and length measurement can then be built upon it.

Thank you for this comment. As you will have seen, this is actually the same concern and suggestion as reviewer 1. As described above (in response to reviewer 1, point 2), we have taken the time to analyse the degraded lunar images. We also now assume a very conservative 1 pixel error ($\sim 50\%$ error) on the track width measurements in our bearing capacity estimates.

Minor comments:

Sup Fig 2: "tracks 1, 2, and 3 are indicated with the orange arrows", the arrows look purple to me!

Fixed, thank you.

Sup Fig 4: Why should the measurement error decrease when degrading to a lower resolution e.g., in Track 2 between 5 m and 6 m, and Track 3 between 5.5 m and 6.5 m?

The width measurements, of the order of 2-3 pixels, are highly sensitive and from a certain resolution the track edges become so indistinct that when placing the markers, we may (by accident) land closer to the original measurement than at the previous resolution step. Globally, the error increases with decreasing resolution.

Sup Fig 8: Can't see the separate dotted/dashed lines. I only see 7 lines which can either be dashed or dotted.

Fixed, thank you.

Methods

1) Use either GM or μ throughout the text.

Fixed, thank you. We now use GM everywhere.

REVIEWER COMMENTS

Reviewer #1 (Remarks to the Author):

Dear authors,

thank you for thoroughly addressing all of my concerns. I went through the revised manuscript and carefully considered your revisions, adding a total of 13 new comments to the attached .pdf file. Overall, I do not have any remaining major concerns, but one minor concern, which was triggered by your revisions and the newly added topographic information. I hope those additional comments are useful to you.

Minor concern – why are two boulders moving up-slope?

Two of the boulder tracks/lineaments appear to lead up-slope, which appears counter-intuitive. I understand that the gravity field favors a displacement of boulders towards the equator (and up a local slope), but I wonder whether that force is strong enough to move boulders of a given size and mass up a slope of up to $\sim 45^\circ$. I would recommend adding a back-of-the-envelope calculation to the manuscript, discussing a) whether that up-slope movement is realistic at all, given a given boulder mass/size, the tracks'/lineaments' specific topography, and the gravity potential delta and b) what this up-slope movement could tell us about the material properties (e.g., at what boulder mass becomes up-slope movement along the two tracks impossible?).

Do those two tracks/lineaments feature terminal boulders? Or is there any other way one could recognize the displacement direction? Maybe those two tracks formed by boulders moving towards the poles, triggered by an energetic event? Ultimately, for the bearing capacity estimate it does not matter why boulders moved and in what direction – however, a thorough geological discussion of those processes would significantly help to build trust in the derived bearing capacity estimates.

Thank you and best regards, Valentin Bickel

Reviewer #2 (Remarks to the Author):

The appreciate the efforts taken by the authors to modify the manuscript significantly and, add new analysis and results. The authors have provided the response to all the comments/suggestions given by the reviewers. However, I still have apprehensions regarding the identification and measurements of the boulder tracks. I would think at least 3 pixels are required to discern the dark-bright gradation of the streak width. I am still not convinced that the streak measured by the authors were made by the boulders.

Other comments I have provided below:

1) How are you sure that these are boulders? When I see Supplementary figure 3, I just see some streaks. When you see a low resolution image of a crater wall, some streaks or lineaments can be observed, where some mass has gone down from higher elevation to lower elevation. How can you surely say that it is boulder and not some mass wasting? The absence of boulders at the end of track again raises a question on the boulders vs simple mass wasting event.

2) Were there any other boulders found which were not in the latitude range of -24 - 16 degrees? And if not, are there any specific reasons for unavailability of boulder tracks in higher latitude range? How long the boulder tracks can survive on the surface of a rubble-pile asteroid?

Lines 91-92: You have written that you see the boulders at the end of the tracks (for some tracks). Can you highlight the boulders at the end of these tracks? Can it be some other features such as mass wasting?

Lines 340-342: Different triggers of boulder falls can result into significant variation in the estimated values of parameters. Could have been discussed in a little more detail.

We are pleased that the reviewers appreciated our efforts to thoroughly revise the manuscript and address each of their concerns. We have replied to their additional comments in blue below.

Reviewer #1 (Remarks to the Author):

Dear authors,

thank you for thoroughly addressing all of my concerns. I went through the revised manuscript and carefully considered your revisions, adding a total of 13 new comments to the attached .pdf file. Overall, I do not have any remaining major concerns, but one minor concern, which was triggered by your revisions and the newly added topographic information.

I hope those additional comments are useful to you.

Many thanks. We reply to your individual comments below.

Minor concern – why are two boulders moving up-slope?

Two of the boulder tracks/lineaments appear to lead up-slope, which appears counter-intuitive. I understand that the gravity field favors a displacement of boulders towards the equator (and up a local slope), but I wonder whether that force is strong enough to move boulders of a given size and mass up a slope of up to $\sim 45^\circ$. I would recommend adding a back-of-the-envelope calculation to the manuscript, discussing a) whether that up-slope movement is realistic at all, given a given boulder mass/size, the tracks'/lineaments' specific topography, and the gravity potential delta and b) what this up-slope movement could tell us about the material properties (e.g., at what boulder mass becomes up-slope movement along the two tracks impossible?).

To improve the clarity of the explanations we have removed all reference to the 'slope' and 'elevation' in this section of the paper. Indeed the only slope that is important to consider is the gravitational slope. We now discuss the gravitational potential only. When we consider the motion in terms of the gravitational potential (Sup. Fig. 1), the boulders are simply moving from a higher to a lower state of potential energy (no specific conditions are required). Additionally, the apparent changes in the elevation may simply be due to small local uncertainties in the shape model so this avoids making conclusions based on data with possibly large uncertainties.

Here is the updated text:

The change in gravitational potential along each of the selected tracks is reported in Fig 2. All 9 boulder tracks exhibit a lower gravitational potential at the lower latitude end of the track (towards the equator) indicating that the boulders would have moved from the higher to lower latitudes in order to minimise their gravitational potential energy.

Do those two tracks/lineaments feature terminal boulders?

No, terminal boulders are not visible in the tracks that had an apparent increase in elevation.

Or is there any other way one could recognize the displacement direction? Maybe those two tracks formed by boulders moving towards the poles, triggered by an energetic event?

It is of course possible that boulders are moving in the opposite direction (from low to high latitudes), but such a motion would be much more difficult to produce. The boulders would have to have enough energy to move up the gravitational potential well, rather than moving naturally down into the potential well. We suggest that the simplest interpretation (of boulders moving from a higher to lower potential energy configuration) is the most likely. This interpretation, as discussed in the introduction, is also consistent with our knowledge of asteroid surfaces and previous observations on other bodies.

Ultimately, for the bearing capacity estimate it does not matter why boulders moved and in what direction – however, a thorough geological discussion of those processes would significantly help to build trust in the derived bearing capacity estimates.

We have tried to improve the language in the paper (removing the notion of the slope and elevation and talking only about gravitational potential). We hope that this, combined with the detailed presentation in the introduction as to why such motion is expected, improve the clarity of these explanations.

Comments from the pdf:

consider replacing "measure" with "estimate"
Replaced.

i think i might have said that before, but a comparison to comet 67P would be interesting, too
consider including comet 67P

In the introduction, we have included a reference to the observations of tracks formed by bouncing boulders on comet 67P (Vincent et al., 2019). However, we have not provided a detailed analysis of boulder tracks on comet 67P. This is out of scope of this work and will be the focus of a separate paper that is in preparation.

If boulders can move up-slope, the existence of slopes as an argument for boulder displacement and cause of the lineaments loses some value.

As demonstrated by the gravitational potential analysis, there is a gravitational slope towards the equator for all of the boulder tracks. However, we have now changed the wording of this section to refer only to the gravitational potential, to avoid confusion with the terms 'slopes' and 'elevation'. In the section of the paper where we calculate the boulder velocities, we now clarify that we are using the gravitational slope (as defined in Barnouin et al. 2024).

shear strength of the soil, not the foundation?
Corrected, thank you.

Only "at" limit equilibrium; before the limit is reached the boulder keeps sinking
Corrected, thank you.

given that this is very curious (at least to me), would you be able to provide a back-of-the-envelope calculation of how heavy the boulders could be to still allow that "anomalous" displacement under the currently known gravity field? and compare those values with the material properties you are currently using for all other computations? I feel some additional thoughts/discussion about this up-slope displacement would add value
See discussion above.

it's curious that the two boulders that move up-slope feature the lowest delta gravitational potential values. one might ask whether the delta is large enough to move the boulders 10 to 15 meters up-slope? on another note, wouldn't that observation enable a constraint of the mass of the boulders?

See discussion above. We have now changed the wording of this section to refer only to the gravitational potential, to avoid confusion with the terms 'slopes' and 'elevation'.

an "equator-facing slope"
This sentence has been deleted.

Thank you and best regards, Valentin Bickel

Reviewer #2 (Remarks to the Author):

The appreciate the efforts taken by the authors to modify the manuscript significantly and, add new analysis and results. The authors have provided the response to all the comments/suggestions given by the reviewers. However, I still have apprehensions regarding the identification and measurements of the boulder tracks. I would think at least 3 pixels are required to discern the dark-bright gradation of the streak width. I am still not convinced that the streak measured by the authors were made by the boulders.

We fully accept our analyses are limited by the image resolution and have tried to make this clear in the paper. Until the arrival of the Hera mission, we cannot have improved images. However, we have followed all of the reviewers' recommendations to quantify the errors induced by the low resolution (Methods 3). This has demonstrated that track width measurements are possible, albeit with a larger error, at the image resolution available.

In addition, in order to be extremely conservative, the Monte Carlo estimates of the bearing capacity used to derive the main results of the paper assume an error of 1 pixel in the track width (equivalent to 50% of the track width). This is much larger than the error estimated from the degraded lunar images (Methods 3) ensuring that the estimates account for the low resolution images.

With respect to the comments about not being convinced that the streaks are made by boulders, we agree that without higher resolution images we cannot be fully certain. However, as outlined in the paper, there are many convincing reasons to believe that the tracks are indeed formed by boulders (see main paper for references):

- The tracks have an appearance similar to boulder tracks on the Moon
- The tracks are parallel and all directed along the gravitational potential slope from the poles to the equator
- The movement of material from the poles to the equator on asteroid surfaces is well accepted and has also been directly observed on other bodies
- The tracks occur in the region of the smooth material on Didymos i.e. the region that is most likely to form tracks
- The tracks do not appear to show an increasing width, indicating that boulder motion is more likely to have caused the features than general avalanching/mass wasting that may be associated with a debris apron.

To reflect this, in the Discussion we now state: *The Hera mission will provide high resolution (0.5-2 m/px) images of the global surface of Didymos. This will provide confirmation that the observed features are indeed tracks formed by boulders. The Hera images will also lead to improved measurements of the tracks investigated here, and perhaps also of the associated boulders in some cases.*

We have also now added the word 'suspected' to the abstract i.e. *images of suspected boulder tracks.*

Other comments I have provided below:

1)How are you sure that these are boulders? When I see Supplementary figure 3, I just see some streaks. When you see a low resolution image of a crater wall, same streaks or lineaments can be observed, where some mass has gone down from higher elevation to lower elevation. How you can surely say that it is boulder and not some mass wasting? The absence of boulders at the end of track again raises a question on the boulders vs simple mass wasting event.

Based on all of the arguments outlined above (and also presented in the paper) there is good reason to believe that the tracks were formed by boulders. The occurrence of these plausible boulder tracks is also discussed in Barnoun et al. (2024) - see figure below.

Figure 8d from Barnouin et al. (2024): Didymos factor of safety (FS) for surface cohesion, $C = 0.5 \text{ Pa}$, assuming a 10 m-thick unconsolidated layer with friction angle $=35^\circ$. Values of $FS < 1$ imply the surface region is prone to failure. The identified boulder tracks in our paper are in the proximity of one of the source regions indicated by white arrows.

We have modified the language in the discussion and abstract to highlight this uncertainty. The absence of visible boulders at the end of all of the tracks could be due to the fact that the boulders have reached enough velocity to be lifted off the surface. We have shown (Sup. Fig. 11) that for different parameter combinations, this can occur. As the leading theory for the formation of Dimorphos is material shed from Didymos' equator that has accumulated in orbit to form the binary (see introduction), this possibility does not seem unreasonable.

2) Were there any other boulders found which were not in the latitude range of -24 - 16 degrees? And if not, are there any specific reasons for unavailability of boulder tracks in higher latitude range? How long the boulder tracks can survive on the surface of a rubble-pile asteroid?

In Pajola et al., (2024), 169 boulders were identified on Didymos from approximately 40°S to 40°N latitude (and 45°W to 45°E longitude). As mentioned above, it seems logical for the tracks to appear only in the lower latitude region; this is the region exhibiting the smoother material on the surface Didymos i.e. the region that is most likely to form tracks (see figure below). In rougher/blockier terrain at higher latitudes, there may be boulder/mass motion, but this is less likely to leave visible tracks.

Figure 2c from Barnouin et al. (2024): The geological units of Didymos

Crater size-frequency analysis places the surface age of Didymos at 12.5 Myrs (Barnouin et al., 2024). Dating of the particular region exhibiting the boulder tracks was not possible with the DART DRACO images, but may be possible with Hera. The long term evolution and/or degradation of the boulder tracks is out of the scope of this current paper.

Lines 91-92: You have written that you see the boulders at the end of the tracks (for some tracks). Can you highlight the boulders at the end of these tracks? Can it be some other features such as mass wasting?

We have indicated some possible boulders with white arrows in Fig. 1a. See discussion above about why we believe that boulder motion is a plausible explanation for these features.

Lines 340-342: Different triggers of boulder falls can result into significant variation in the estimated values of parameters. Could have been discussed in a little more detail.

The following text has been added to the Discussion: *If the initial boulder velocities are larger this could result in the boulder motion moving out of the quasi-static regime invalidating the use of the Terzaghi equation. However, the initial velocity must also be lower than the lift-off velocity; if the initial boulder velocity is close to or above the lift-off velocity then the boulders would lose contact with the surface before forming the tracks.*

REVIEWERS' COMMENTS

Reviewer #1 (Remarks to the Author):

Dear authors,

Thank you for making another round of revisions. I do not have any remaining, noteworthy concerns. I left a couple of comments in the attached .pdf file, which might turn out to be useful (or not).

One note: there seems to be an issue with the article file PDF; the file ends at supplementary figure 4, cropping the following figures, reference section, et cetera. I assume this is a formatting/rendering issue? I assume there were no changes to the material on those cropped pages? If there were, it would make sense for the reviewers to get another chance to check this modified material again. If there were no changes to the content beyond SUP Fig. 4, I do not need to see the manuscript again.

Thank you and Godspeed for Hera,
Valentin Bickel

Reviewer #2 (Remarks to the Author):

The authors have taken substantial efforts to analyze and modify the manuscript according to the comments provide by the reviewers. Looking at all the explanations provided by the authors and changes incorporated in the manuscript, I feel that the manuscript can be recommended for the publication. I would just like to ask the authors to go through their equations once more and provide the definition of all the symbols and format them properly.

Thank you for your very helpful and constructive comments all throughout the reviews that improved the quality of our paper. We addressed your last comments in the final version of the manuscript and below is a point-by-point response to those.

Reviewer #1 (Remarks to the Author):

Dear authors,

Thank you for making another round of revisions. I do not have any remaining, noteworthy concerns. I left a couple of comments in the attached .pdf file, which might turn out to be useful (or not).

Thank you, we took note of your suggestions related to the wording and reformulated some sentences. We also revised the numbering and format of the equations to be suitable for publication.

One note: there seems to be an issue with the article file PDF; the file ends at supplementary figure 4, cropping the following figures, reference section, et cetera. I assume this is a formatting/rendering issue? I assume there were no changes to the material on those cropped pages? If there were, it would make sense for the reviewers to get another chance to check this modified material again. If there were no changes to the content beyond SUP Fig. 4, I do not need to see the manuscript again.

Sorry, there was indeed an issue with the PDF file that we did not notice. We ensure that there were no modifications to the content that follows, supplementary figures and references.

Thank you and Godspeed for Hera,
Valentin Bickel

Reviewer #2 (Remarks to the Author):

The authors have taken substantial efforts to analyze and modify the manuscript according to the comments provide by the reviewers. Looking at all the explanations provided by the authors and changes incorporated in the manuscript, I feel that the manuscript can be recommended for the publication. I would just like to ask the authors to go through their equations once more and provide the definition of all the symbols and format them properly.

Thank you, we went through our equations and revised their formatting, also making sure that all the symbols we use are well defined.